# Features of Media Multitasking in School-Age Children

**DOI:** 10.3390/bs9120130

**Published:** 2019-11-27

**Authors:** Galina Soldatova, Svetlana Chigarkova, Anna Dreneva

**Affiliations:** 1Department of Personality Psychology, Faculty of Psychology, Lomonosov Moscow State University, 11/9 Mokhovaya, 125009 Moscow, Russia; soldatova_gv@psy.msu.ru; 2Department of Social Psychology, Moscow Institute of Psychoanalysis; 34/14 Kutuzovsky prospect, 121170 Moscow, Russia; 3Department of Methodological Psychology, Faculty of Psychology, Lomonosov Moscow State University, 11/9 Mokhovaya, 125009 Moscow, Russia; annadreneva@msupsy.ru

**Keywords:** media multitasking, user activity, academic performance, executive functions, children and adolescents

## Abstract

The paper addresses the phenomenon of media multitasking that is being widely spread among children and adolescents in the context of digital socialization. The previous research has revealed its strong connection with cognitive control, executive functions, and academic performance, yet the specificity and efficacy of media multitasking performance, especially among children while they carry out usual activities, remains insufficiently studied. A quasi-experimental study, including digital tasks of various types on a computer and smartphone, the dots task for executive functions, and a socio-psychological questionnaire, was conducted with the participants of three age groups: 7–10, 11–13, and 14–16 years old (N = 154). The results indicate that media multitasking is connected not with sex, but age; the older the participants are, the more likely they tend to work in a multitasking mode. Furthermore, preference for multitasking has been found to be positively related to higher user activity. Although the total task performance rate is insignificantly lower in the multitasking group as compared to the non-multitasking one, a significant negative effect of media multitasking on total performance time was revealed. The results of the study that indicate a strong connection of media multitasking with the intensity of Internet usage, cognitive functions, and performance time, suggest its considerable role in social and cognitive functioning of children and adolescents.

## 1. Introduction

Modern society is determined by the increasing role of diversity, complexity and uncertainty, which are mediated by various factors, including digital technology implementation in everyday life. In the process of adapting to such sociocultural space and using the diverse repertoire of cultural e-tools, people widely switch to multitasking mode, including media multitasking. This raises the question of its effectiveness, limitations and impact on cognitive functions, especially among the developing age groups—children and adolescents. Being at the forefront of digital socialization processes, the younger generation is becoming the most active users of digital devices. Their lifestyle is determined by the use of the wide range of digital devices, switching between different online environments and also between online and offline, which creates conditions for media multitasking mode [1]. According to the recent study, about 60% of adolescents aged between 10 and 15 report being regularly media multitasked [2].

Recent studies identify the phenomenon of media multitasking from three perspectives: (1) the simultaneous use of several media technology tools; (2) the combination of using media and traditional sources of information; (3) the combination of using offline and virtual activities [3,4,5,6,7,8,9,10]. While analyzing existing research on multitasking, we found a number of works considering the correlation of media multitasking with various cognitive processes in adolescents and youth: Volume and shifting of attention, low stability, and concentration of attention, as well as cognitive control and executive functions [1,9,10,11,12,13,14,15]. Most of the studies have revealed the negative effect of media multitasking on cognitive processes, yet a number of papers have presented more positive conclusions. Heavier media multitasking has been associated with worse performance on fluid intelligence measures such as Raven’s Progressive Matrices [13], on demanding working memory tasks [9,15,16,17], and on task switching [15]. On the contrary, the study of American youth shows that media multitaskers are characterized by a developed ability to switch attention and cognitive control [18]. Other studies have demonstrated that respondents who media multitask have a broader scope of attention [11,12]. One of the few studies of adolescents’ executive functions and media multitasking mode showed an association between heavier media multitasking and better inhibitory control [3]. 

One of the directions of media multitasking research is the study of its relationship with academic performance. The majority of these studies demonstrate that media multitasking is negatively related to academic performance such as academic outcomes (i.e., grades), study-related behaviors and attitudes, and perceived academic learning [19]. However, the observed negative relationships or effects were often small to moderate and not always significant.

Demographics (gender and age) may also be related to media multitasking. In two studies the mean media multitasking scores for adolescent samples were markedly lower than the mean scores for young adult samples [3,16]. Another study found that young people are better at media multitasking than adults [20]. Moreover, several studies have shown that women are better at media multitasking [5,20]. However, many other studies have either not found or not reported gender effects of media multitasking [15,21]. 

The methodology of media multitasking research has some weaknesses. Self-assessment (Media Multitasking Index) is the most common tool for measuring levels of media multitasking [13,17,21]. Quasi-experimental and experimental schemes are rarely used in media multitasking research. Media multitasking studies are more often conducted on a youth or student sample, and data on children and adolescents are presented in few studies [3,14,16,22,23]. Most studies use the extreme group method, which examines the differences between two groups of "light" and "heavy" multitaskers and does not include non-multitaskers [11,13,16,17,18,21].

Although increasing digitalization and rapid changes in everyday life are pressing issues, there is a lack of research in the area of media multitasking and principally studying this phenomenon among children and adolescents during their daily digital activity. Hence, the aim of our study was to investigate the peculiarities of media multitasking in children of different age groups while using digital devices. 

The main hypotheses of the study are the following: H_1_) A preferred level of media multitasking is connected with sex and age; H_2_) a preferred level of media multitasking is connected with user activity and academic performance; H_3_) level of media multitasking is connected with executive functions and cognitive control. 

## 2. Methods

The study was conducted according to the principles of the Declaration of Helsinki and was approved by the local ethics committee. All the parents and legal representatives of the children gave a written informed consent before the study.

### 2.1. Participants

The sample of the study consisted of children aged between 7 and 16 (N = 154) with the experience of using digital devices and their parents (N = 154) living in Moscow. We recruited the participants in a way to balance the sample by sex (77 boys and 77 girls) and age (50 primary school children aged 7–11, 60 secondary school children aged 11–13, and 44 secondary school adolescents aged 14–16). The children studied in different Moscow schools of similar social context, demography and performance characteristics. About 80% of the parents had tertiary education, and the same amount of families were middle-income. 

### 2.2. Design of the Trial

For this study, a methodical complex was developed and tested. It is comprised of the following parts: (1) Socio-psychological questionnaire on sociodemographic characteristics, academic performance, and intensity of online user activity; (2) quasi-experimental scheme of multitasking mode when using a computer and smartphone; (3) the Hearts and Flowers dots task in the modification of T.A. Akhutina, A.A. Korneyev, and A.N. Gusev [24]. 

In the context of our study, we operationalized the concept of media multitasking as shifting among different types of digital tasks or their parallel performing. Unlike other studies that often use self-assessment tools (Media Multitasking Index) [15,17,22], we also developed and used a quasi-experimental scheme to reproduce the conditions of everyday media multitasking. The scheme comprised a number of tasks reproducing typical students’ activities. The tasks included: (1) Searching online for the definition of an unknown word; (2) solving several arithmetic tasks and tasks on rearranging syllables in several words in Google forms; (3) reading an online text; (4) watching a short video; (5) answering the questions from three messages sent to the mobile phone during the experiment. For each age group, the tasks were slightly modified to correspond to the age level of complexity.

To study executive functions and cognitive control, the dots task was used. This task required to press the response button in reaction to the signal. This task was performed on a computer and included three blocks with gradual increasing of complexity: A block of 20 congruent trials (with all responses on the same side to a dot (heart) appeared); a block of 20 incongruent trials (with all responses on the opposite side opposite to a dot (flower) appeared; a mixed block of 20 trials combining both congruent and opposite probes. The main parameters were the quality of performance (number of correct answers) and average response time.

### 2.3. Procedure

The participants were sitting with their own smartphones at the interviewer’s computers. The participants used their own smartphones, which could differ in some technical characteristics. The main requirements were: Internet access and the devices’ habitualness. The participants received printed instruction with a tasks list and the following oral instructions from the interviewer: “Now you will have to complete several tasks. They are written on this paper. Please, read and remember what you will have to do. You will not have the opportunity to look at it again. You should complete them as quickly as possible in any order”. Then, the printed instruction was taken. Four windows were opened on the screen. Every 2 min the interviewer sent the message with the question. During the experiment, five different music segments were played in the background (croaking of frogs, drums, guitar, chorus from a children’s song, the sound of the surf). After completing the tasks, the interviewer asked several questions about the meaning of the text, a few details about the video, music, and then, after filling out the questionnaire, the definition of the word searched on the Internet.

The study was conducted at home in families in the form of an individual interview with each child while the parents filled out the socio-psychological questionnaire. The data were collected in 2017 and 2018.

### 2.4. Data Processing

The data were processed using IBM SPSS Statistics 23. P-value < 0.05 was considered as statistically significant.

## 3. Results

### 3.1. Division into Groups According to the Level of Media Multitasking, Sex, and Age

Groups were divided by three main factors: Sex, age group, and media multitasking level (MML). The last parameter was determined on the basis of such attention properties as shifting and distribution. In our study, it was represented through the number of additional tries of solving each task. For each respondent, all additional tries were summed up for all the tasks. Then, using the expert assessment method, we distinguished three media multitasking levels: “non-multitasking” (the absence of additional tries), “low multitasking” (1–2 additional tries), and “high multitasking” (3 and more additional tries—maximum was 12). Thus, the sample was divided into 66 non-multitaskers, 63 low multitaskers, and 25 high multitaskers.

### 3.2. Relationship of MML, Sex, and Age Group

To investigate H_1_, we performed a chi-squared test. No significant relationship between MML and sex was found while the age group had a strong relationship with MML: The older were the participants, the higher level of multitasking they showed (χ^2^ = 19.971, df = 154, p = 0.001) (Table 1). Notably the least number of low and high multitaskers was shown in the middle group.

### 3.3. Relationship between MML and User Activity

The sample was divided into four groups by user activity on weekdays: Low activity (less than 1 h per day), normal activity (1–3 h per day), high activity (4–5 h per day), and hyperconnectivity (more than 6 h per day). To check H_2_, we used a chi-squared test and observed the significant relationship between user activity group and MML (χ^2^ = 15.482, df = 147, p =0.017). In the group of high multitaskers, half of the participants demonstrated high user activity or hyperconnectivity. Every third fell into the group of normal user activity, and only every sixth was in the group of low activity. At the same time, a half of low multitaskers fell into the group of normal user activity, and every fifth was characterized by high user activity or hyperconnectivity. Among non-multitaskers, a half was in the group of low user activity, and every fifth was in the hyperconnectivity group (Table 2).

### 3.4. Relationship between MML and Academic Performance

No significant relationship between MML and academic performance was observed. However, it was found that almost two thirds of high multitaskers’ academic performance were below average, and one third had good performance while in the low and non-multitasking groups, an inverse proportion was found (Table 3).

### 3.5. The Influence of MML on Total Quality Score

The performance efficacy was revealed to be independent of MML (Figure 1). Although non-parametrical comparison of non-multitaskers and low multitaskers using the Mann–Whitney test shows trend differences (U = 1706.5, p = 0.078).

The analysis of MML and age group influence on total performance time using median criterion showed a significant impact of both factors (χ^2^ = 15.512, df = 152, p < 0.000 for age group, and χ^2^ = 7.685, df = 152, p = 0.021 for multitasking level) (Figure 2).

One of the parameters presumably connected with multitasking was the strategy of watching videos: Full watching, fast-forwarding, and watching only the beginning. The significant relationship between MML and the strategy of watching videos was revealed (χ^2^ = 13.12, df = 152, p = 0.011) (Table 4): High multitaskers were more likely to fast-forward the video rather than to watch it fully (40% of high multitaskers against only 14.3% of low multitaskers and 9.4% of non-multitaskers).

According to H_3_, the analysis of age group and MML influence on the results of the dots test indicated a significant effect only on the most complex task. Two-way ANOVA with two independent variables (age and multitasking group) and one dependent (number of correct answers in the dots test) revealed a significant influence of the age factor (F = 4.068, p = 0.019) and a pronounced tendency for the MML (F = 2.986, p = 0.054) (Figure 3). 

The results indicated that the non-multitaskers (of all the age groups) turned out to be the most effective in the most difficult task comprising such executive functions as inhibition and shifting. In the youngest group, the high multitaskers coped worse than the low ones. However, in the middle and older groups, the level of performance between them became identical (Figure 3).

## 4. Discussion and Conclusions

The research findings reveal that age, unlike sex, is significantly related to media multitasking: The older the children are, the more they are likely to have a higher level of media multitasking. Similar trends were identified in another study on adolescents [23] that has shown that the predictors of media multitasking are age, impulsivity, and digitalization of a family. The last parameter is indirectly related to the level of user activity, which has been analyzed in this work: The more actively the children use digital devices, the more they tend to have higher level of multitasking; and the higher the level of user activity is, the "heavier" is multitasking. We have also confirmed that user activity increases with age.

Additionally obtained was a negative relationship between the level of media multitasking and academic performance, which is consistent with the results of several studies [3,11]. Thus, among the high multitaskers, almost two thirds demonstrate lower academic performance, while for the non-multitaskers and low multitaskers, higher performance is more common. This may indicate that high multitasking is a non-optimal strategy for academic studies.

Despite the fact that media multitasking has not affected the total quality score, it turned out to be a significant factor for the parameters of total performance time. It was observed that the multitaskers of both types, as compared to the non-multitaskers, have performed the tasks practically at the same quality level. However, they have spent more time. This allows to suggest that the distribution of attention and the ability to focus on several tasks are not always voluntarily regulated and are often reduced to a constant and chaotic shifting of attention among several current tasks. Nevertheless, such multitasking mode may indicate that the participants who incline to multitask are adapting to the complex process of distributing attention among several tasks. However, the results have revealed this strategy to be less optimal as opposed to the sequential and completing strategy. This is proved by the fact that high multitaskers’ preference to fast-forward or watch partially a video does not lead to an expectedly faster performance.

Moreover, the multitaskers of both levels have demonstrated worse performance in the mixed dots task, which integrates both congruent and incongruent probes. Furthermore, in the youngest group, the high multitaskers have coped worse than the low ones while in the older age groups this difference has been eliminated.

Thus, discussion showed that executive functions and cognitive control are inversely connected with the level of media multitasking, which has also been shown in the study on adolescents aged 11–15 [3]. In this work, it is particularly presented in the younger group as compared to the older one, which can be explained by the development factor and maturation of brain structures [25].

## 5. Limitations and Future Prospects

This paper presents the results based on a selected part of the data concerning academic performance, user activity, and executive functions of children with different media multitasking modes. The further investigations will be based on the other part of the data through several directions. Firstly, it is the level and characteristics of family digitalization that should be investigated thoroughly as it can determine the propensity to a different level of media multitasking. Secondly, it is the data of neuropsychological research that will allow to relate the multitasking level with such functions as memory, attention, speech, reasoning, and visual-motor integration. Thirdly, it is the further operationalization of media multitasking criteria that can provide a more detailed investigation of this phenomenon.

## Figures and Tables

**Figure 1 behavsci-09-00130-f001:**
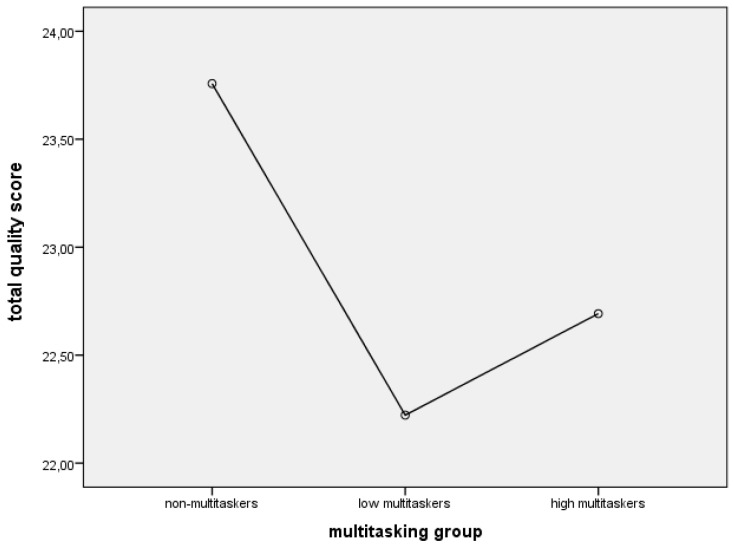
The influence of MML on total quality score.

**Figure 2 behavsci-09-00130-f002:**
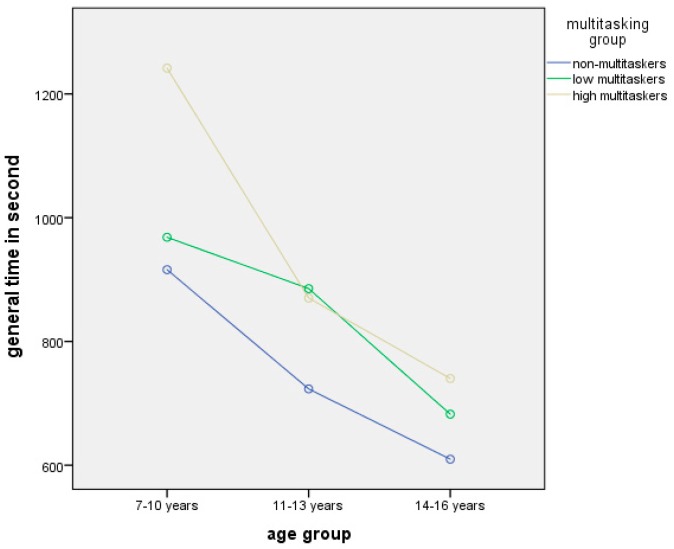
The influence of MML and age group on total performance time.

**Figure 3 behavsci-09-00130-f003:**
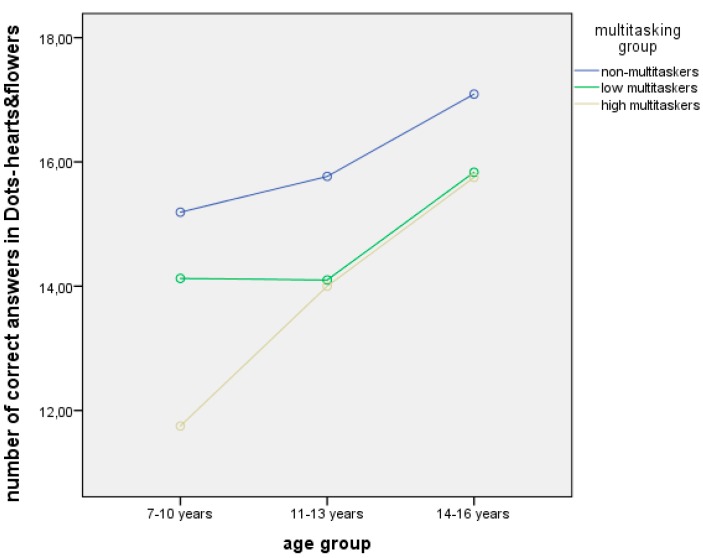
The influence of MML and age group on the results of the mixed dots task.

**Table 1 behavsci-09-00130-t001:** Relationship between age and media multitasking level (MML).

	Non-Multitaskers (%)	Low Multitaskers (%)	High Multitaskers (%)
7–10 years old	42	50	8
11–13 years old	56.7	33.3	10
14–16 years old	25	40.9	34.1

**Table 2 behavsci-09-00130-t002:** Relationship between user activity and MML.

	Low Activity (%)	Normal Activity (%)	High Activity (%)	Hyperconnectivity (%)
Non-multitaskers	42.6	21.3	16.4	19.7
Low multitaskers	29	50	11.3	9.7
High multitaskers	16.7	33.3	25	25

**Table 3 behavsci-09-00130-t003:** Relationship between academic performance and MML.

	Excellent (%)	Good (%)	Normal (%)	Bad (%)
Non-multitaskers	4.9	59	32.8	3.3
Low multitaskers	6.5	51.6	38.7	3.2
High multitaskers	8.3	33.3	58.3	0

**Table 4 behavsci-09-00130-t004:** Relationship between the strategy of watching videos and MML.

	Watched Fully	Fast-Forward	Only Beginning
Non-multitaskers	70.3	9.4	20.3
Low multitaskers	66.7	14.3	19
High multitaskers	52	40	17.8

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
