# Peer review of "Features of Media Multitasking in School-Age Children"

_behavsci, 2019, doi:10.3390/bs9120130_

Round 1
Reviewer 1 Report
INTRODUCTION
---------------
While not an expert in the field, I have some concerns about the theoretical background, related works, and motivation, which are briefly discussed in the "Introduction" section:
- I would like to read more about the theoretical background of media multitasking, cognition, and behavior. Maybe a better framing focusing on the aim of the paper would help.
- I would like to see more discussion about what positive and negative effects have been identified in previous studies (Lines 45-47). How do they build the motivation of the paper?
- I cannot see a rationale behind the hypotheses and how they are built. I would expect the authors to provide a clear motivation and present the knowledge gap (e.g., by referencing other studies), and thus, justify the hypotheses.
METHODOLOGY
----------------
While the methodology is clearly described, I miss some information that hinders the replicability of the study.
- I assumed that the balanced groups were not randomly formed (e.g., having 77 boys and 77 girls). Therefore, I would expect the authors to discuss in more detail the recruitment process and the formation strategy they followed.
- What procedure did the authors follow regarding the ethic? Was there any consent form? Was it filled by both parents and children?
- Did both parents fill the forms?
- I would expect a more detailed description of the dots task.
RESULTS
----------
- Please connect the subsections with the hypotheses investigated. For example in section 3.2, the authors could add "To investigate H1, we performed a xxx statistical analysis".
- Since the authors perform statistical analysis, they could develop null hypotheses that they retain or reject after the analysis of the results. The null hypotheses should be based on the hypotheses discussed in the Introduction section.
- Please clearly mention the dependent and independent variables (and covariates if any) for each statistical test.
- Please mention what statistical tests you used.
MISC.
------
- There are some typos throughout the text, therefore, careful proofreading is suggested.
- Please consider using "They" instead of "He or she".
Reviewer 2 Report
Overall, I consider that this is a promising study.
I mention that my comments are aimed most of all at the media philosophy and theory that underpin the study, and not at the sociological research.
To begin with, the article needs language editing. Particular attention should be given to the use of articles and tenses.
This can be claimed subjective, but I do not agree with the authors' opening statement:
"Modern society is determined by the increasing role of diversity, complexity and uncertainty, mediated by the active digital technology implementation in everyday life."
To begin with, modernity is a bygone age and digital technology and media are rather characteristic of (what should rather be called) postmodernity. More important, still, is that, while digital media are critical to contemporary social organization and lifestyles, they do not fully "determine" society altogether. Surely, there must also be other factors and other types of media that contribute to shaping society.
Regarding methods: are the levels of engagement with digital devices of the subjects comparable? Did they come from different schools? Can we know the demography, social context and performance of the schools? In what language was the experiment carried? Did the participants have different types of mobile/smart phones?
I do not understand clearly the points made in "Limitations and Future Prospects". Could the authors please explain more clearly? Why are they publishing preliminary results? Will it not be redundant to publish a later a study comprehensive of more data? Are they using a selected sample of the data for this paper?
The authors can consider the following article, in light of their argument on multitasking in regard to age, which does not relate multitasking competences with age precisely, but with digital nativity:
Scolari CA, Masanet MJ, Guerrero-Pico M, Establés MJ. 2018. Transmedia literacy in the new media ecology: Teens’ transmedia skills and informal learning strategies. El Profesional de la Informacion, 27(4): 801-812.
Author Response
1. I see the article is clear, and organized well. within the context of the article, the result of the experiment is logic. However, Although the Phenomenon of media multitasking is worthy to be analyzed by several approaches, the introduction provided by the author(s) is concise much more than it should be. a section of two or three paragraphs to analyze the related literature is inevitable in order to demonstrate the significance of the current studies so that the author can illustrate how this article represents a step forward.
1. Thank you, we added several passages with the related literature on different aspects of multitasking.

Reviewer 3 Report
I see the article is clear, and organized well. within the context of the article, the result of the experiment is logic. However, Although the Phenomenon of media multitasking is worthy to be analyzed by several approaches, the introduction provided by the author(s) is concise much more than it should be. a section of two or three paragraphs to analyze the related literature is inevitable in order to demonstrate the significance of the current studies so that the author can illustrate how this article represents a step forward.
Best regards
Author Response

(The authors gave the same response as above.)

Round 2
Reviewer 1 Report
Thank for your comments. Please make sure to include the ethical statement.